

# Ephemeral learning – Augmenting triggers
# with online-trained normalizing flows

**Anja Butter[1,2], Sascha Diefenbacher[3⋆], Gregor Kasieczka[3],**
**Benjamin Nachman[4,5], Tilman Plehn[1], David Shih[6] and Ramon Winterhalder[7]**

**1** Institut für Theoretische Physik, Universität Heidelberg, Germany
**2** LPNHE, Sorbonne Université, Université de Paris, CNRS/IN2P3, Paris, France
**3** Institut für Experimentalphysik, Universität Hamburg, Germany
**4** Physics Division, Lawrence Berkeley National Laboratory, Berkeley, CA, USA
**5** Berkeley Institute for Data Science, University of California, Berkeley, CA, USA
**6** NHETC, Department of Physics & Astronomy, Rutgers University, Piscataway, NJ USA
**7** Centre for Cosmology, Particle Physics and Phenomenology (CP3),
Université catholique de Louvain, Belgium

⋆ sascha.daniel.diefenbacher@uni-hamburg.de

## Abstract

The large data rates at the LHC require an online trigger system to select relevant colli­sions. Rather than compressing individual events, we propose to compress an entire data set at once. We use a normalizing flow as a deep generative model to learn the probabil­ity density of the data online. The events are then represented by the generative neural network and can be inspected offline for anomalies or used for other analysis purposes. We demonstrate our new approach for a toy model and a correlation-enhanced bump hunt.

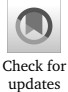

# 1 Introduction

The ATLAS and CMS experiments at the Large Hadron Collider (LHC) produce data rates around 40 terabytes per second and per experiment [1,2], a number that will increase further for the high-luminosity upgrades [3,4]. These rates are far too large to record all events, so these experiments use triggers to quickly select potentially interesting collisions, while discarding the rest [5–8]. The first two trigger stages are a hardware-based low-level trigger, selecting events with $\mu$s-level latency, and a software-based high-level trigger with 100 ms-level latency. After these two trigger stages, some interesting event classes, such as events with one highly-energetic jet, still have too high rates to be stored. They are recorded using *prescale* factors, essentially a random selection of events to be saved. An additional strategy to exploit events which cannot be triggered on systematically is data scouting, or trigger-level analysis [9–12]. Through fast online algorithms, parts of the reconstruction are performed at trigger level, and significantly smaller, reconstructed physics objects are stored instead of the entire raw event. This physics-inspired compression increases the number of available events dramatically, with the caveat that the raw events will not be available for offline analyses.

Using machine learning (ML) to increase the trigger efficiency is a long-established idea [13], and simple neural networks for jet tagging have been used, for example, in the CMS high-level trigger [14]. The advent of ML-compatible field-programmable gate arrays (FPGAs) has opened new possibilities for employing such classification networks even at the low-level trigger [15–21]. ML-inference on FPGAs is making rapid progress, but the training of e.g. graph-based networks on such devices is still an active area of research. At the same time, the available resources limit the size and therefor complexity of possible ML models.

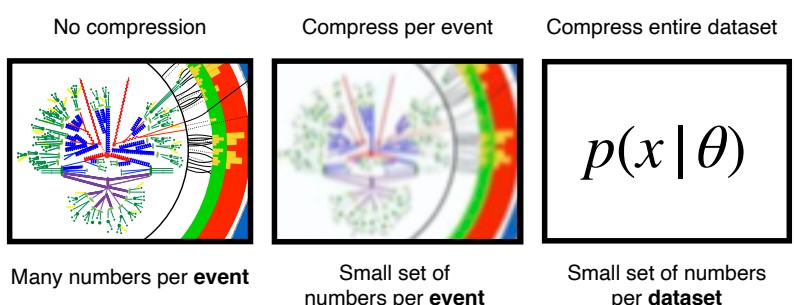

Figure 1: Illustration of data compression at the LHC. Most analyses are performed offline, based on entire events and lossless compression (left). Data scouting employs lossy compression per event (center). Our method compresses an entire data set by learning a generative model for events $x$ in terms of network parameters $\theta$ (right).

We propose a new strategy, complementary to current trigger strategies and related methods, where instead of saving individual events, an online-trained generative ML-model learns the underlying structure of the data. The advantage of our strategy, illustrated in Fig. 1, is its fixed memory and storage footprint. While in a traditional trigger setup more events always

require more storage, the size of the generative model is determined by the number of parameters. Additional data increases the accuracy of these parameters at fixed memory size, until the capacity of the model is reached. In practice, we envision an online generative model to augment data taking at the HLT level[†] and act as a scouting tool in regions currently swamped by background. However, a sufficiently optimized version of this approach could transform data taking by instead learning the overall distribution of data without the need for triggers altogether.

The viability of our novel approach rests on the assumption that the relevant physics of LHC collisions can be described, statistically, by far fewer parameters than are necessary to record an entire event. An intuitive example is a set of $N$ Gaussian random numbers. Recording the entire data set requires $N$ numbers, but the sample mean and variance are sufficient statistics, so that only storing them contains all the information about the entire data set. Using generative models for data compression is also an established method [22–26]. However, it usually means encoding a given data point into a more storage-efficient representation. This is different from our proposal which aims to encode the full underlying distribution in the network parameters [27] and to represent all aspects of the LHC training data in the generative network. Modulo limits of expressivity, the online-trained network output can, in principle, replace the training data completely.

This paper is structured as follows: In Sec. 2 we discuss the challenges of running an online trained generative model and strategies by which these can be overcome. In Sec. 3 we perform a first proof-of-concept study using a simple 1-dimensional data set. Section 4 expands this test to a standard benchmark data set. We present our conclusions in Sec. 5.

## 2  Online training

Trigger systems work through a chain of consecutively stricter requirements. Directly following the sensor measurements are the low-level or level-1 (L1) triggers. They have to make decisions fast enough to keep up with the rate of incoming collisions, in our case 40 MHz, so they are hardware-based and at most perform low-complexity reconstruction. The passing events then reach the high-level trigger (HLT). The reduced data rate output by the L1 triggers, 100 kHz for ATLAS and CMS, allows for a software-based HLT running on a dedicated server farm. Its primary purpose is to reduce the event rate to the point where everything can be stored for offline analysis. In some cases it can be beneficial to have an HLT channel with low thresholds, such that the number of triggered events is still too large to be stored completely. In such cases one randomly decides which event is stored or discarded. The chance to store an event and hence the data reduction is given by a tuneable prescale suppression factor.

We propose a new, ML-based scouting strategy in the form of an online-trained generative model. A generative model trained on events, which are not triggered and stored, can extract interesting information without saving the events. We therefore consider augmenting the HLT as a first possible application of the new technique. The proposed workflow, also shown in Fig. 2, is

- (online) Train a generative model on all incoming events;

- (offline) Use the trained model to generate data;

- (offline) Analyze this generated data for indications of new physics;

- (offline) If an interesting feature appears, adjust the trigger to take data accordingly;

---

[†]as training (as opposed to inference) models on FPGA hardware deployed at earlier trigger stages is currently not possible

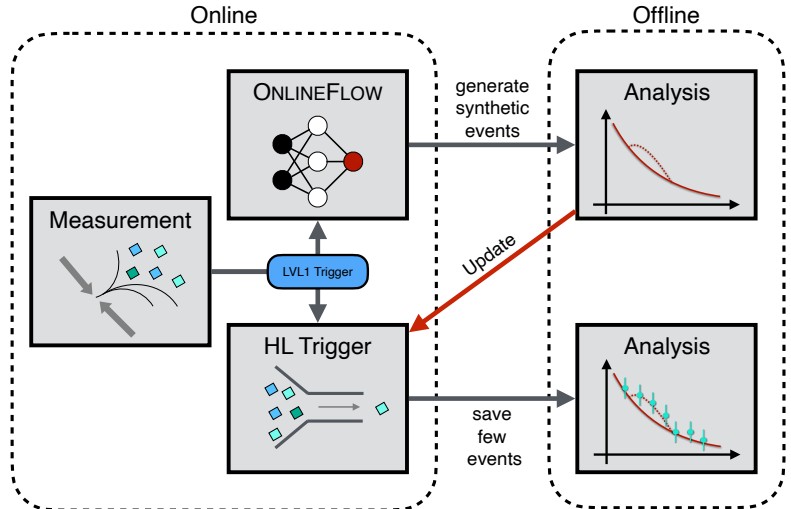

Figure 2: Illustration of the proposed workflow. First, we train a generative model on all incoming events (online). Then, we use the trained model to generate data and analyze the generated data for signs of new physics (offline). If necessary, we adjust the trigger to take new data accordingly (online) and analyze that data (offline).

- (online) Collect data with new triggers;
- (offline) Statistically analyze that recorded data.

This online model is compatible with most current trigger setups[†]. As a first step we assume operation in the HLT, but the concept is also applicable to the first trigger stage. The workflow can also be extended to measurements rather than searches.

While our idea is not tied to specific generative models, normalizing flows (NF) [28–31] are especially well suited due to their stable training. This allows us to train our ONLINEFLOW without stopping criterion, a property well suited for training online. Furthermore, NFs have been shown to precisely learn complex distributions in particle physics [32–42]. The statistical benefits of using generative models are discussed in Ref. [43], for a discussion of training-related uncertainties using Bayesian normalizing flows see Refs. [44, 45].

The properties of online training, specifically seeing every event independently and only once, are in tension with training generative models. Such models perform best when they have the option to look at data points more than once. Additionally, processing several events at the same time should allow the model to train significantly faster through the use of GPU-based parallelization and stochastic gradient descent. This is why we follow a hybrid approach: incoming events are collected in a buffer with size $N_{\text{buff}}$. Once this buffer is full, it is passed to the network, which processes the information in batches of size $N_{\text{batch}}$. This process is iterated over $N_{\text{iter}}$ times. After this, the buffer is discarded and replaced by the next buffer. We visualize this scheme in Fig. 3. In addition to aiding the network training, this hybrid training also decouples the network training rate from the data rate, as we can continuously adapt $N_{\text{iter}}$ to ensure the network is done with the current buffer by the time the next is filled. Additional technical details, including the estimation of uncertainties, of our approach are discussed in the context of the examples presented below.

---

[†]It may also be possible to discover new physics directly with the generated data, in contrast to adjusting the triggers. However, we take a more conservative perspective here.

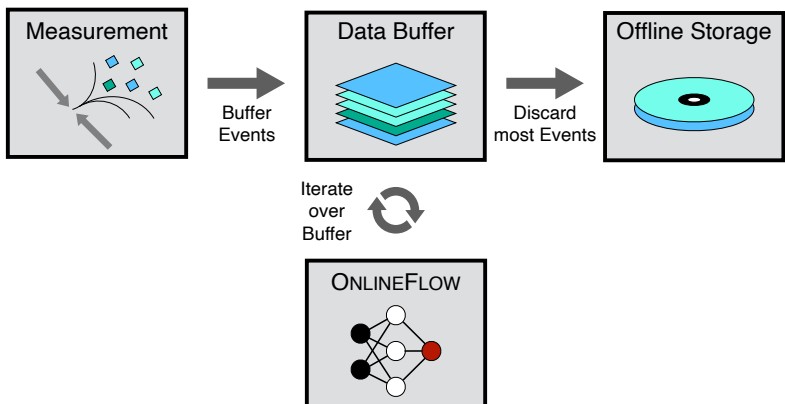

Figure 3: Illustration of our test of the online training. We collect events (left) into a buffer (top center). In the classic prescale approach most events are discarded, only a small fraction is saved for offline analysis (right). Our ONLINEFLOW (bottom center) is trained online on the data in the buffer and iterated until the buffer is replaced.

# 3 Parametric example

We first illustrate our strategy for a 1-dimensional parametric example. While in practice it would be straightforward to store at least a histogram for any given 1-dimensional observable, this scenario still allows us to explore how generative training and subsequent statistical analysis approaches need to be modified for the ephemeral learning task.

## 3.1 Data, model, training

The 1-dimensional data is inspired by a typical invariant mass spectrum with a resonance. In a sample with $N$ events, or points, every event is randomly assigned to be either signal or background with a probability of $\lambda$ or $1 - \lambda$, respectively. On average, this gives $\lambda \times N$ signal and $(1 - \lambda) \times N$ background events. The values $x$ of the signal and background events are drawn from their respective distributions,

$$p_B(x) = \frac{1}{b} e^{-bx} \quad \text{and} \quad p_S(x) = \frac{1}{\sigma \sqrt{2\pi}} e^{-\frac{1}{2}(\frac{x-\mu}{\sigma})^2} . \tag{1}$$

We use $b = 1$, $\mu = 1$, $\sigma = 0.04$ and $\lambda = 0.005$ and show the truth distribution in Fig. 4.

As our generative model we choose a masked auto-regressive flow (MAF) [46]. It comprises five MADE [47] blocks, each with two fully connected layers with 32 nodes, resulting in 7560 network weights. To aid the flow in learning the sharp threshold, we train on the logarithm ($\log x$) of each 1-dimensional event. Furthermore, since NFs such as the MAF are not designed for 1-dimensional inputs, we add three dummy dimensions drawn from a normal distribution, quadrupling the total number of input and output dimensions to four. The MAF model is implemented using PYTORCH [48] and trained using the ADAM optimizer [49] with a learning rate of $10^{-5}$ and a batch size of $N_{\text{batch}} = 250$. Finally, we use a buffer size of $N_{\text{buff}} = 10,000$ and $N_{\text{iter}} = 100$ iterations per buffer. Our training sample includes 5M events altogether, stored for evaluation purposes.

One challenge with this model is a bias towards the more recent buffers, for which the network is optimized on last. To cure this, we use stochastic weight averaging [50]. During training, we keep track of the running average of the model weights. Once the network is done with a given buffer, the average weights are updated, scaled by the number of average-weight updates so far. At the end, we use the averaged weights for generation.

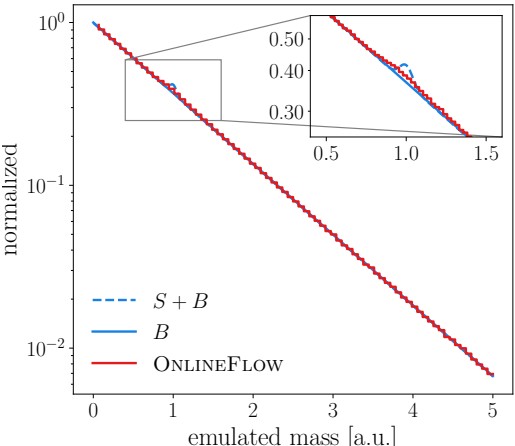

Figure 4: Illustration of our 1-dimensional, exponentially falling, mass spectrum. *B* denotes background events, *S* is the signal, following the truth distributions of Eq.(1). The OnlineFlow histogram shows 10M events generated after training on $S + B$.

In Fig. 4 we show how OnlineFlow, trained on signal plus background attempt to reproduce the signal mass peak. While the width of the peak is not correctly described we do see a noticeable abundance. As the goal is in finding evidences of new physics in a subsequent offline analysis of the OnlineFlow data, this abundance may still, however, be sufficient for our purpose even if the peak width is mismodeled. To estimate the statistical uncertainty[§] (described in more detail below), needed to compute *p*-values, we train 20 flows in parallel. As training samples for the statistical uncertainty, we use bootstrapping. Creating classical bootstrapped ensembles is not possible because we do not have the full dataset to resample from with replacement. Instead, we use an online-compatible version whereby each event in each bootstrapped ensemble is given a weight that is Poisson distributed with unit mean. These weights are independent for each flow in the ensemble and are kept constant as the flow iterates over one buffer. We leave the further exploration of this ad hoc solution and alternative methods such as Bayesian flows [44, 51, 52] to the future.

## 3.2 Classical bump hunt benchmark

Our goal is to compare how well a potential signal can be extracted from flow-generated events, vs. a range of classical offline analyses, consisting of standard bump hunts on the training data reduced by different levels of prescale triggers. First, we describe our procedure for the latter.

We use SciPy [53] to fit a background model to the mass histogram of $N_{\text{pre}} = N_{\text{data}}/f_{\text{pre}}$ events, where $N_{\text{data}} = 5 \times 10^6$ and $f_{\text{pre}}$ is the prescale factor. We find that the following background model fits the data well:

$$p(x) = \alpha \, e^{-\beta x + \gamma x^2 + \delta x^3 + \epsilon x^4 + \zeta x^5} \,, \tag{2}$$

with best fit parameters that depend on $f_{\text{pre}}$.

We supply this background model to BumpHunter [54] to identify the most likely signal region. Our BumpHunter-setup scans the emulated mass ranging from 0 to 5, divided into 50

---

[§]Assuming that the variations in the output from fixed inputs and the stochastic nature of the network initialization and training are negligible compared to the data statistical uncertainty. If this is not the case, one could reduce these with further ensembling.

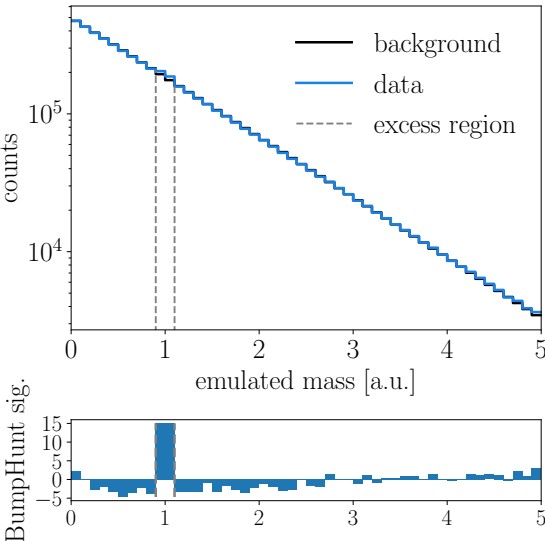

Figure 5: Example of the BUMPHUNTER used on the training data, based on the background model of Eq.(2). Dotted lines indicate the upper and lower bounds of the signal region. The lower panel shows the significance.

bins, with a minimal signal window size of two bins and maximal window size of six. Given the data, BUMPHUNTER extracts a lower and an upper bound on the most likely bump position and defines our signal region. We then extract the local significance as

$$\text{significance} = \frac{\mathcal{O} - B}{\sqrt{B}} \equiv \frac{S}{\sqrt{B}}, \tag{3}$$

where $B$ is the number of background events predicted in the signal region, and $S$ is defined as the number of observed events $\mathcal{O}$ minus background model. As the prescale factor is increased, we expect the local statistical significance of the bump hunt to decrease as $1/\sqrt{f_{\text{pre}}}$. We illustrate an example of the classical analysis in Fig. 5, corresponding to $f_{\text{pre}} = 1$ (i.e. using the full 5M training data). We see that the background model agrees well with the data over the entire range, except for the identified signal region (and the surrounding regions, which must compensate for the excess).

### 3.3 ONLINEFLOW performance

For ONLINEFLOW, we train (in the online fashion described above) on the full 5M events, and use an ensemble of networks to generate 100M events, combined into one large set. This large number is used to make the statistical uncertainty from sampling negligible compared with other sources of uncertainty. We again fit the same background model, Eq.(2), to the mass histogram of these 100M events, and find the best fit parameters to be

$$
\begin{aligned}
\alpha &= 1.0099194(5), & \beta &= 1.020952(15), & \gamma &= 0.03216(4), \\
\delta &= -0.017450(14), & \epsilon &= 0.003913(1), & \zeta &= -0.00031395(1).
\end{aligned} \tag{4}
$$

While the background model is trained on a large sample of flow-generated data, its $\chi^2$ for the smaller set of training events and a smaller set of flow-generated events is excellent ($\chi^2/\text{dof} \approx 1$) and consistent with each other. We have also checked that changing the analytic form of the background model has little effect on our results.

As for the classical analysis, the best-fit background model is then input to BUMPHUNTER in order to identify the most likely signal region. To reduce the chance of mistaking imperfect

network trainings for a signal, we randomly split our model ensemble into two equal parts, other splits being possible as well (such as $k$-folding; see e.g., Refs. [55,56]). The first ensemble of ten networks defines the signal region using BUMPHUNTER. Given the signal region, we then use the second ensemble of ten networks to compare the number of generated events in the signal region to the predicted background.

To estimate the significance of the signal encoded in the ONLINEFLOW, it is a bit more subtle than just using Eq.(3), since we are essentially taking the statistical uncertainty of the generated events to be zero by generating 100M of them. Instead, the statistical uncertainty on the training data is translated into a systematic uncertainty in the generated events. To quantify this, we use the second network ensemble to compute bootstrap statistics in the usual way. First, combining all flow-generated events in the signal region, $\mathcal{O}$, and the event count from the respective background fit, $B$, gives us the total signal and background in the signal region and the corresponding uncertainties

$$B = \frac{2}{N_{\text{ens}}} \sum_i^{N_{\text{ens}}/2} B_i \,, \qquad\qquad \delta_B = \sqrt{\frac{2}{N_{\text{ens}}}} \, \sigma(B) \,,$$

$$\mathcal{O} = \frac{2}{N_{\text{ens}}} \sum_i^{N_{\text{ens}}/2} \mathcal{O}_i \,, \qquad\qquad \delta_{\mathcal{O}} = \sqrt{\frac{2}{N_{\text{ens}}}} \, \sigma(\mathcal{O}) \,, \qquad (5)$$

where $\sigma(X)$ is the standard deviation of $X$ over the ensemble and, in our case, $N_{\text{ens}}/2 = 10$. The signal rate and uncertainty,

$$S = \mathcal{O} - B \,, \qquad \delta_S^2 = \delta_{\mathcal{O}}^2 + \delta_B^2 \,, \qquad (6)$$

define the signal significance

$$\text{significance} = \frac{S}{\sqrt{\delta_S^2 + (\sqrt{B})^2}} \,. \qquad (7)$$

The contribution $\sqrt{B}$ represents the flow statistical uncertainty from the finite amount of generated samples. It will usually be negligible compared to the data statistical uncertainty, $\delta_S$, but we still include it in our calculation for consistency.

Because for our parametric example we can save the training data, as well as the weights of all models after each buffer, we can track the signal significance during the online training. In the left panel of Fig. 6 we compare the significance of the ONLINEFLOW and the complete training data. As expected, the data-derived significance scales with the square root of the number of events. The ONLINEFLOW significance initially rises at a similar rate, and then increases at a slower rate. Asymptotically, the significance may saturate if the network approaches the limits of its expressiveness.

In the right panel of Fig. 6, we compare the performance of ONLINEFLOW vs the classical bump hunt with different prescale factors $f_{\text{pre}}$. We see that, as expected, the significance of the latter decreases as $1/\sqrt{f_{\text{pre}}}$. Meanwhile the significance returned by ONLINEFLOW is constant since the network is always trained on the full data. Above $f_{\text{pre}} \approx 4$, we see that ONLINEFLOW starts to outperform the classical approach.

Finally, we need to check how susceptible our ONLINEFLOW setup is to fake signals. We run our setup with the same parameters as before, but for zero signal fraction. The result is shown in Fig. 7. We see that there is a larger error margin for the ONLINEFLOW significance, owing to the larger fluctuations between individual ONLINEFLOW training, however the average fake rate for the flow stays well below the signal significance we achieve for the 0.5% signal contamination. The average also stays consistent with the fake rate of the classical bump hunt.

The precise behavior for intermediate signal rates presents an interesting question that exceeds the scope of this work, but would warrant further investigation.

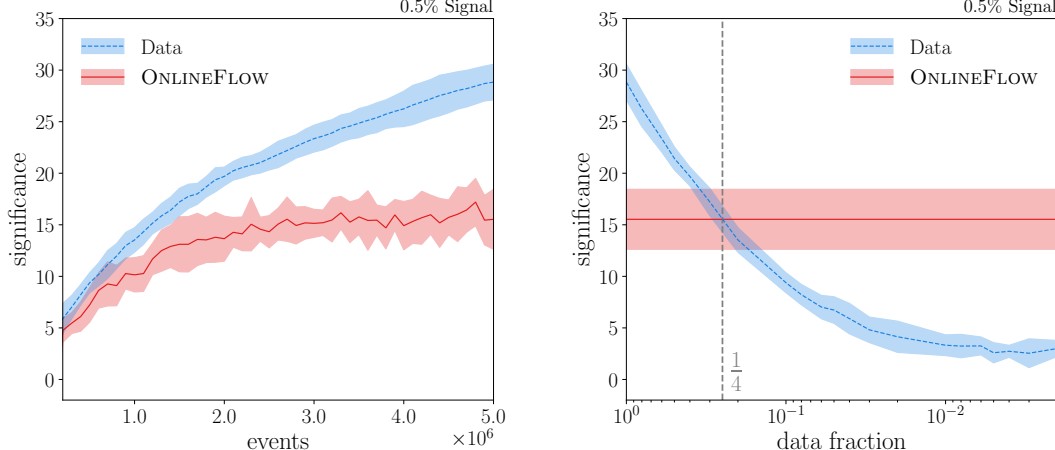

Figure 6: Left: signal significance as a function of the amount of training data for the classical approach, based on all training data, and the ONLINEFLOW. Right: signal significance as a function of the prescale factor. A prescale factor of one corresponds to $500 \times 10^4$ events. The shaded regions estimate the uncertainty based on five executions of the experimental setup. The dotted grey line indicates the crossover point at a datafaction of $\frac{1}{4}$, which corresponds to a prescale factor of 4.

## 4 LHCO dataset

After illustrating our novel approach for the 1-dimensional toy model, we move to the more realistic, higher dimensional LHC Olympics anomaly challenge R&D dataset [57].

### 4.1 Data, model, training

The dataset comprises a background of dijet events including a new-physics signal. This signal, which we assume to contribute 1% of our events, originates from a $W'$ decaying into two heavy particles, which in turn decay into quarks,

$$W' \to X(\to qq)Y(\to qq). \tag{8}$$

The respective particle masses are $m_{W'} = 3.5$ TeV, $m_X = 500$ GeV, and $m_Y = 100$ GeV. All events are generated using PYTHIA8 [58] and DELPHES3.4.1 [59–61]. The jets are clustered using FASTJET [62] with the anti-$k_T$ algorithm [63] using $R = 1$. Finally, all events are required to have at least one jet with $p_T > 1.2$ TeV.

While this dataset features high mass resonances that are not perfectly in line with the intended application range of ONLINEFLOW, we feel that the proven and well known nature of the LHCO data, as well as its availability make up for this shortcoming.

The same input format used for the anomaly detection [32, 40, 64] is also used for the ONLINEFLOW. Specifically, there are five input features, the dijet mass, the mass of the leading jet, the mass difference between the leading and sub-leading jets, and the two $n$-subjettiness ratios [65, 66],

$$\left\{ m_{jj}, m_1, m_2 - m_1, \tau_{21}^{(1)}, \tau_{21}^{(2)} \right\}. \tag{9}$$

All observables except for $m_{jj}$ are subjet observables and at most weakly correlated with $m_{jj}$. We show distributions of all observables in Fig. 8, for the training data and the ONLINEFLOW

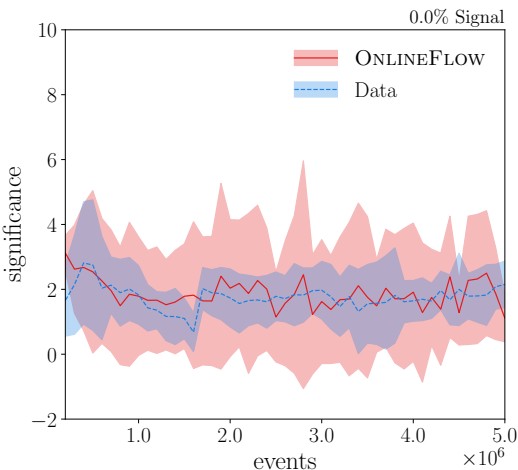

Figure 7: Fake signal significance as a function of the amount of training data for the classical approach, based on all training data, and the ONLINEFLOW. The analysis is identical to that of Fig. 6, but with zero signal fraction.

output. We also show a 10-fold enhanced signal, relative to the 1% signal rate we will use for our actual analysis, to illustrate the narrow kinematic patterns of the $W'$ resonance.

The LHCO version of the ONLINEFLOW network is slightly modified compared to the parametric setup to accommodate a 10-dimensional input. These comprise five features and five additional noise dimensions, the additional noise was found to increase the performance, although no systematic scan over this hyperparameter was performed. The number of MADE blocks is now 10, and the number of nodes in the fully connected layers is quadrupled to 128. The buffer size is still 10,000, and the number of iterations per buffer is increased to 1,000. Learning rate and optimizer are identical to those used before.

For an assumed signal rate of 1% we split a total of around 300k LHCO events into 80% training, 10% evaluation, and 10% validation data. The limited evaluation data is further supplemented with additional signal and background events, to smooth out the ROC and SIC curves. This gives us a new set of 330k evaluation events, roughly equally split between signal and background, but only to present our results.

The red lines in Fig. 8 demonstrate the flow's ability to reproduce the five input features. The leading jet mass and the mass difference are learned well, as are the $n$-subjettiness ratios. The invariant dijet mass distributions shows some deviations at the sharp boundaries, a typical effect for neural networks which can be cured using a range of standard methods [45].

## 4.2 CWoLa benchmark

Just like for the parametric example, we need to determine how well the ONLINEFLOW captures the signal features for an anomaly detection setup. As a simple benchmark we choose the Classification Without Labels (CWoLa) [55, 56, 67] setup, implemented following Ref. [40]. In the CWoLa framework, a classifier is trained to distinguish between two samples with different relative amounts of signal. For the LHCO dataset these two samples are the signal region defined in terms of the dijet invariant mass and indicated in Fig. 8,

$$m_{jj} = m_{W'} \pm 200 \text{ GeV} = 3.3 \ldots 3.7 \text{ TeV}, \tag{10}$$

and the control regions away from the $W'$-peak. While this definition of a signal region has no effect on the training of the ONLINEFLOW, it implies that we cannot include $m_{jj}$ as a training

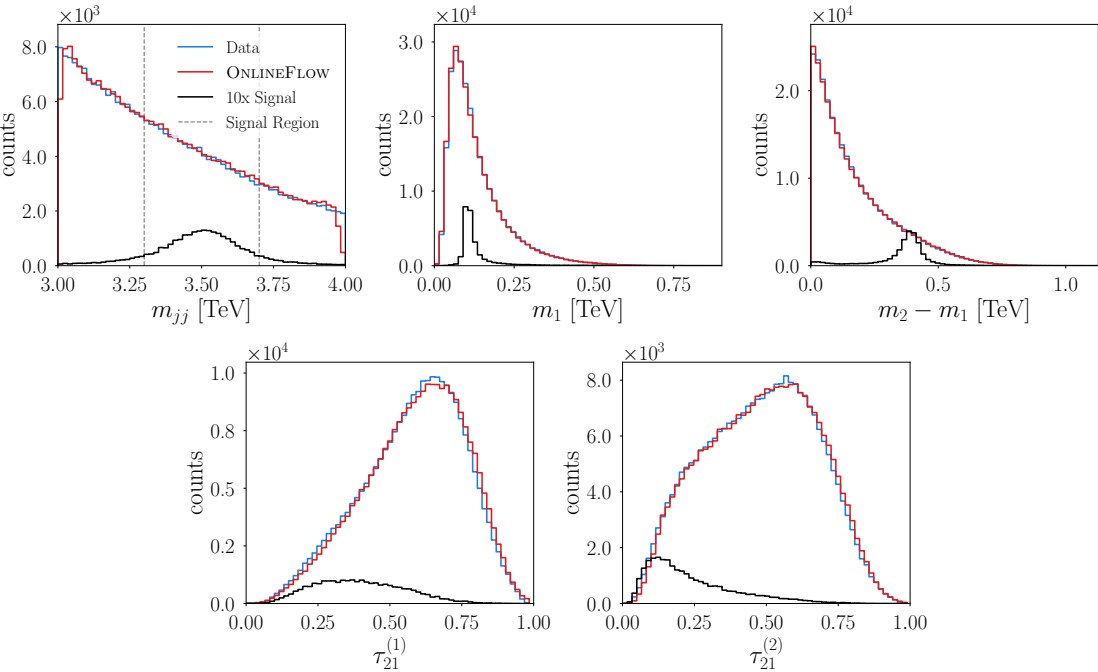

Figure 8: Observables for the LHCO data set, as listed in Eq.(9). We show the original training data, with 1% signal contamination, and the data generated by the flow. The signal region in $m_{jj}$ is indicated by dotted lines.

variable for the CWoLa network. In a realistic anomaly search one would use a sliding mass window, but since we are only interested in benchmarking the ONLINEFLOW we assume a signal region around the resonance.

As the signal and control regions should only differ in the amount of potential signal, the classifier learns to distinguish signal from background while separating the two regions. We can define the likelihood ratio for a given event $x$ to be signal as

$$R_{\text{CWoLa}}(x) = \frac{p(x|\text{SR})}{p(x|\text{CR})}, \tag{11}$$

where $p(x|\text{SR})$ and $p(x|\text{CR})$ are the classifier outputs for signal and control regions. By scanning different thresholds on this ratio we can enrich the relative amount of signal-like events.

Our CWoLa classification network is implemented using PYTORCH [48] and consists of three fully connected layers, each with 64 nodes. The training uses a binary cross entropy loss, the ADAM [49] optimizer with a learning rate of $10^{-3}$, and runs for 100 epochs. As the signal and control region contain an unequal amount of data, the two classes are reweighted to correct for this imbalance. The final classifier comprises an ensemble of the ten network states with the lowest validation loss during training.

To benchmark the signal extraction of the ONLINEFLOW, we train CWoLa on the LHCO training data. It contains 240k events and is identical to the data used to train the ONLINE-FLOW. This training is aided by a 30k validation set. To determine how much information the ONLINEFLOW captures we repeat the CWoLa benchmark training with 50%, 20%, 10%, and 5% of the LHCO training and validation sets.

### 4.3 ONLINEFLOW performance

Considering the positive results from our simple toy model, we now compare the CWoLa results based on the ONLINEFLOW and on the LHCO training data. The relevant numbers of

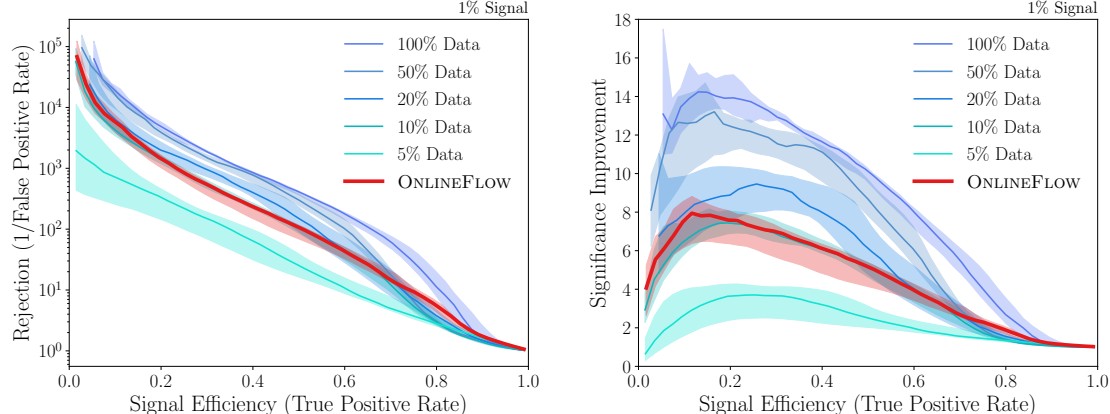

Figure 9: ROC (left) and significance improvement (right) for the CWoLa benchmark approach, based on a decreasing amount of data, compared with the ONLINEFLOW. The signal fraction is 1%. Vertical order of the Data lines corresponds to their order in the legend.

merit, namely the ROC curve and the signal improvement $\epsilon_S/\sqrt{\epsilon_B}$, are shown in Fig. 9. For the standard CWoLa approach, trained on the LHCO data, the smaller training samples correspond to prescale factors of 2, 5, 10, and 20. The shaded regions indicate the one-sigma range from repeating the CWoLa analysis ten times. We see that, for instance for a constant signal efficiency, the background rejection drops increasingly rapidly for smaller training samples. This illustrates how larger prescale factors seriously inhibit the reach of searches for new physics in non-trivial kinematic regions.

To determine the power of the ONLINEFLOW we then train the CWoLa network on 500k events generated from the ONLINEFLOW, with an additional 62500 ONLINEFLOW events serving as the validation set. This mirrors the split into training-validation-test data of the LHCO data. In both panels of Fig. 9 we can now compare the ONLINEFLOW results to the different prescalings and find that it performs similarly to 10% of the training data. In a setting where one has to work with a trigger fraction of less then 10%, one could benefit from the ONLINEFLOW setup.

While the CWoLa results show that the ONLINEFLOW does not only encode features represented in the input variables, but also describes correlations directly, it remains to be shown that its performance is stable when we decouple the main features more and more from the input variables. This happens when we train the generative networks on low-level event representation, challenging the network both in expressivity and reliability. In line with the conclusions from Fig. 6 this might, for instance, require a larger network and adjustments to the building blocks of the normalizing flow and the bijectional training.

## 5 Conclusions

Data rates of modern particle colliders are a serious challenge for analysis pipelines. In terms of data compression, triggered offline analyses use lossless data recording per event, but at the price of a huge loss in deciding which event should be recorded. Trigger-level analyses accept losses in the individual event information, to be able to analyze significantly more events. Our strategy is inspired by the statistical nature of LHC measurements and aims at analyzing as many events as possible, but accepting a potential loss of information on the event sample level.

For this purpose, we propose to train a generative neural network, specifically a normalizing flow (ONLINEFLOW), to learn and represent LHC events for offline analyses.

First, we have studied the performance of this ONLINEFLOW compression for a 1-dimensional parametric example. We mimic a narrow bump hunt on an exponentially dropping flat background and compare the significance achieved by classic methods with different prescale factors with the significance based on generated events from the trained ONLINEFLOW. We find that ONLINEFLOW outperforms a classical, offline analysis for prescale factors larger than 3.5, while fake signal significances remain at the level of the classical analysis once the flow is trained properly.

Second, we looked at a more realistic example, specifically a simulated $W'$-signal available as the LHCO R&D dataset. We use the CWoLa method to extract the signal from correlated phase space observables and to define the benchmark based on the training events with a variable prescale factor. Again, we find that ONLINEFLOW outperforms the offline CWoLa method for prescale factors above a certain threshold (in this case $\sim 10$).

Implementing ONLINEFLOW into an existing trigger system will require further work to scale up the networks input dimensionality as well as its expressiveness to handle the more complex data structures of real LHC events. Further, the challenge of integrating the model training into the infrastructure of a real experiment will require further work and exploration. However, regardless of these challenges, we believe the examples demonstrated here serve as a proof of concept for the proposed ONLINEFLOW, warranting further investigation.

# Acknowledgements

The research of AB and TP is supported by the Deutsche Forschungsgemeinschaft (DFG, German Research Foundation) under grant 396021762 – TRR 257 Particle Physics Phenomenology after the Higgs Discovery. AB would like to thank the LPNHE for their hospitality. This work was supported by the Deutsche Forschungsgemeinschaft under Germany's Excellence Strategy EXC 2181/1 - 390900948 (the Heidelberg STRUCTURES Excellence Cluster). SD is funded by the Deutsche Forschungsgemeinschaft under Germany's Excellence Strategy – EXC 2121 Quantum Universe – 390833306. BN is supported by the U.S. Department of Energy, Office of Science under contract DE-AC02- 05CH11231. The work of DS was supported by DOE grant DOE-SC0010008. RW is supported by FRS-FNRS (Belgian National Scientific Research Fund) IISN projects 4.4503.16. This research was supported in part through the Maxwell computational resources operated at Deutsches Elektronen-Synchrotron DESY, Hamburg, Germany.

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
