# Peer review of "Ephemeral Learning -- Augmenting Triggers with Online-Trained Normalizing Flows"

_SciPost Physics, doi:SciPost Phys. 13, 087 (2022)_

## Round 1 · Referee Report · Anonymous (Referee 1) · 2022-4-5

Strengths

  1. The paper presents a novel use of generative neural networks to characterize ensembles of events in a periodic online manner to program a certain class of triggers.

  2. The results of the paper show that the proposed trigger approach is, for scenarios considered in the paper, more efficient than conventional approaches used at the LHC using bump hunts, as well as more recent proposals like offline CWoLa algorithms that relies on data samples without truth labels such as "signal" and "background". The juxtaposition of various problems and tools makes it easy for the reader to quickly see its effectiveness.

  3. The paper considers a parametric toy problem to demonstrate the tool as well as a more realistic physics scenario that the tool can be used for at the LHC. The build-up makes it easy for the reader to understand the tool.

Weaknesses

  1. The paper considers a realistic example wherein the existing conventional cut-threshold approach may suffice. This may be a moot point with some textual clarifications. See question/comment 3.

  2. The tool as potentially being deployable in FPGA/GPU seems difficult with the amount of detail given in the paper. This may be a moot point with some textual/figure clarifications. See questions/comments 8, 14, and 15.

Report

Congratulations to the authors for the interesting proposal. The approach and the accompanying result show important progress in the field. This makes the paper worth publishing.

Some minor revisions/clarifications are requested.

Requested changes

Questions/comments regarding the physics/proposal presented:

  1. The impact the signal contamination has on the training could be made clearer. For example, one might naively guess that the signal is tiny in the beginning and that each update will make the signal significance larger. Here in lies the confusion. If the signal is larger after the update, won’t the subsequent update absorb the signal in the background model? Some clarification may alleviate such concerns.

  2. As a follow-up to the previous point, can/does OnlineFlow account for changing pileup conditions? Assuming that it does this well, to what order-of-magnitude level of S/B can this tool find new physics? Presumably given enough training cycles and updates it can squeeze out the significance given any S/B.

  3. The physics case considered is the LHC Olympics dataset containing a W’ decay filtering on having a jet with pT > 1.2 TeV. This feel like an odd choice to showcase as the threshold for the lowest unprescaled single jet trigger at CMS and ATLAS was much lower at around 500 GeV during Run-2. One can imagine an argument that the specifics of this problem does not matter as it is demonstrating the capability and that for deployment lower pT jets would be targeted. Is that true? Assuming that it is, the reader is left to wonder whether the results of Figure 9 still holds. Some clarification would be appreciated.

  4. The role, rationale, and impact of dummy variables is vague. The appearance of dummy variables in the parametric example seems less controversial since standard optimization tools may not work for for 1d inputs for technical reasons. Three is a mystery here, but not of big concern. However, for the LHCO problem the inputs are doubled with dummy variables. What is going on?

Editorial comments/suggestions approximately in order presented in the paper:

Section 1

  1. The introductory paragraph is confusing as it is trying to accomplish many things. Perhaps it might be helpful to break it up so that each paragraph has a point. For instance, the paper starts rather generally, but transitions quickly into rather technical details such as prescales, and ends with scouting / trigger level analysis. That’s the general comment, now some specific parts.

  2. The introductory sentence gives the impression that the collision rate might be going up, rather than the trigger rate at the same collision rate. Perhaps there is a way to reword.

  3. The presentation of the scouting / trigger level analysis can be better. As it’s presented, even with the word “additionally”, it looks like the main use case of the trigger system at the LHC is scouting/trigger level analysis. The authors may have tried to motivate the paper as soon as possible without actually giving the physics scenario they have in mind, and this caused much of the confusion.

  4. The second paragraph on ML is confusing. It describes the offline use, FPGA use, then GPU use cases. After the FPGA references [15-21], reference [15] is cited alone in the following sentence. What separates [15] from the others that were cited before? Also, “sophisticated networks” is rather subject unless you specify the limitations of existing systems that prevent you from accomplishing something that you have in mind. Lastly the final sentence ends with “this idea [22]” that leaves the reader wondering what it is. What is also lacking in this paragraph is the connection to the next one, because the proposed strategy seems limited to the HLT. There are some hints in the paper that it may be able to be deployed on FPGA/GPU, but lacks details.

  5. The final sentence of the third paragraph is rather general without details. It requires too much work on the reader’s part to try to construct trigger menu, even a toy one with only a few main items, using the proposed method.

  6. The last sentence of the penultimate paragraph is confusing. Isn’t the network determination based on training data?

  7. Would a sketch or example of the idea of “generative ML” be too pedagogical? Perhaps even a simple definition would help.

Section 2

  1. In the first paragraph, it is claimed that L1 trigger algorithms do not perform complex reconstructions. CMS deployed BDT on FPGA during Run-2, which might qualify as complex to some, so this claim seems somewhat subjective. The idea seems to be that L1 is not as complex relative to the HLT, so perhaps it can be reworded.

  2. The bulleted steps is helpful. One part that triggered a question is step 3, where it mentions “indication of new physics”. How can the system distinguish new physics compared to a detector flaw? The authors may know of a paper by A. Pol et al (https://arxiv.org/abs/1808.00911) that discusses detector monitoring using anomaly detection. It would be interesting to know what the authors think of the new physics vs. detector issues ambiguity, even if it is beyond the scope of the paper. If appropriate, please add the citation.

  3. The presentation of where and how OnlineFlow could be deployed could be made more concrete. Only after reading the paper does the reader understand that the tool is to be deployed at HLT-like environments where all of the reconstructed inputs, including sophisticated variables that require tracking information, are made available as inputs. This preprocessing is taken for granted—which may be somewhat reasonable for HLT, although that would depend whether full tracking is needed—and this would not be a given for FPGA/GPU-like setups.

  4. For example, figure 2 is confusing in where OnlineFlow gets its Measurement, which I think is in the HLT-like preprocessing environment. In the current diagram a line connected to the detector. Since the Update arrow is feeding into both LVL1 and HLT, the inclusion of the LVL1 adds to the confusion. It almost feels like the paper would be stronger if it focused on the HLT application for now, with the possibility of FPGA/GPU implementation in the future. This seems to be what is already desired in the paper, but the presentation is a bit confusing.

Sections 3

  1. Section 3.1, after referencing figure 4, states that the OnlineFlow reproduces the peak. OnlineFlow seems to have a weak bump above background, but it is much flatter than the S+B. Is that what you mean? This feels like it did not reproduce the peak. Please clarify.

  2. In section 3.3, it’s not clear to the reader that the parameters given in equation 4 are important enough to document it in the paper. Perhaps it is sufficient to state them in the figure? Also, the relevant figure should be referenced somewhere earlier in section 3.3.

  3. Is there significance of prescale factor ~ 4? It’s not obvious to the reader why 4, if significant. Perhaps it is empirical to the problem at hand. Please clarify.

  4. The final paragraph of section 3 is rather dense and is difficult to follow. Why is the errorband in the OnlineFlow larger, in general, compared to the data? Can this be reduced by sampling more frequently or is it an intrinsic feature of the network as chosen?

Section 4

  1. Figure 9 and others. It is difficult to correspond the color to the legend as is especially for the color challenged reader or a reader with a black-and-white printer. If you wish to keep the current aesthetics, perhaps a comment could be added to state that the order of the legend follows the curves, if it indeed does, or change the order such that this holds.

  2. Another suggestion on the figure is to make the line in the legend much thicker so that the colors show well. More orthogonality in color choices and color intensity would help.

  3. Lastly, regarding figures, it might help to highlight the result for OnlineFlow by choosing, for example, a thicker line.

Section 5

  1. “prove” should be “proof” in the final paragraph.

Acknowledgments

  1. Format of quotes closing Quantum Universe. Various past and present tenses used. Not sure if intentional.

References

  1. Bibliography is in need of serious work. Some examples are given. Incorrect authors (1, 2), missing authors (3, 4), journal abbreviations are inconsistent (Phys. Rev. Lett., JINST, Journal of Instrumentation, Physical Review Letters, etc.), inconsistent capitalizations (cms, tev, fpga, lhc, etc.), incorrect grammar (and et al. in 17), incorrect formatting (et al. in italics and in roman), redundant or missing links (doi is given twice in 32, etc), misformatted subscripts (anti-ktjet, etc.), internal notes remaining in (8, 18), inconsistent use of et al (15 gives ten authors before et al., whereas 22 gives one author).

  • validity: -
  • significance: -
  • originality: -
  • clarity: -
  • formatting: -
  • grammar: -

Author:  Sascha Diefenbacher  on 2022-06-29  [id 2621]

(in reply to Report 1 on 2022-04-05)

We would like to thank the reviewer and we greatly appreciate the detailed and thoughtful feedback and questions. We tried to address them in a way that hopefully makes the paper more understandable. Replies to the individual points follow:

1- Thank you for raising this point, however we are unsure about the source of the confusion. The signal fraction remains constant throughout a single training, as it would be purely defined by the underlying physics. The update indicated in Fig.2 is not intended as an automated, continuous process that takes effect during the training, but rather as an after-the-fact cross check in case any hints of new physics are found in the OnlineFlow scheme.

2- We agree that the exact behavior of the OnlineFlow for various signal rates would be interesting to explore, however we feel this to be beyond the current scope of this work. We added a remark about the behavior for various S/B ratios and how this is an interesting region to explore. As for pileup, the objects that go into the flow would be pileup corrected, so there should be no difference compared to an offline search.

3- Thank you for noting this, while this dataset features high mass resonances that are not perfectly in line with the intended application range of OnlineFlow, we feel that the proven and well known nature of the LHCO data, as well as its availability make up for this.

4- The origin of the three dummy variables stems from doubling the input twice, from 1 to 2 to 4 in total. The addition of the dummy variable in the LHCO setting is indeed not required, however we saw and improvement to the training behavior through their inclusion. This however not systematically explored. We further added more clarification on the origin as use of the dummy variables to the paper

Section 1 5- Thank you for the feedback; we will address each of your detailed comments below

6- Thank you for pointing this out, however the purpose of the first sentence is to say that the collision rate is too high to record all events - there is nothing to do with the trigger there. Therefore, we prefer to keep these sentences as they are. The logical flow here is: (1) collision rate is too high to save everything. (2) Need to save partial information, either by throwing out events, throwing out parts of events, or both. Each method has advantages and disadvantages.

7- Thanks for bringing this to our attention, we changed “additional” to “alternative”

8- We appreciate the feedback and removed the superfluous citation and reworded the section.

9- We clarified the section and hope it is now more understandable

10- While the network is based on the training data, those data are not saved. What we mean by this sentence is that the generative model trained online could be equivalent to saving all of the training data for offline”.

11- While we can understand the point, we do feel that a introduction into generative models would not be the best fit for this paper.

Section 2 12- Thank you for noting this, we reworded the section in question.

13- We understand how this question arises, however the OnlineFlow setup itself does not aim to distinguish between new and known physics. The principled idea is to store a representation of the measured data while using less effective storage space. How this representation of the data is later on treated in specific is beyond the scope of this work, the two anomaly detection methods used (BumpHunt and CWoLa) serve as a non-exhaustive example. The update to the trigger system is not intended to happen in an automated online setting, but rather involves modifying trigger menus after analyzing the representation of the data in an offline setting.

14- We understand the point, however we mention the deployment on HLT as the first proposed deployment level quite early on in this section and we feel that an even earlier mention of this would disrupt the text flow too much.

15- The original figure combined both triggers into one box for the sake of readability, which did leave the specifics of what Input is used for the Online Flow ambiguous. We have since modified the figure to hopefully better represent the intended application.

Sections 3 16- As the goal is in finding evidence of new physics in a subsequent offline analysis of the \OF data, any abundance, even if it does not perfectly model the signal peak, may still be sufficient to find new physics. We have added a sentence to this section detailing this and further elaborating on the signal peak agreement.

17- We do feel the parameters are important as they allow for comparison between the fit result on the OnlineFlow data and the true underlying function parameters. Stating them in the figure unfortunately resulted in a difficult to read figure. Unfortunately we are not sure which figure is referred to here, as in our eye the fit parameters would not seem appropriate or relevant for the significance comparison in Fig.6. Further Fig.5, which does show a fit, does not use OnlineFlow data, but the analysis performed on the training data, so adding the parameters here does not seem intuitive either.

18- The prescale factor of 4 is not inherently significant, however it is the prescale factor at which the OnlineFlow significance and data significance crossover. To further help that point we specifically indicated the crossover point in the relevant figure.

19- Thank you for pointing this out, we agree that the final section is very dense and have added additional elaboration to remedy this.

Section 4 20- The vertical order of the lines in the legend does indeed correspond to their order in the plot. We added this point to the figure caption.

21- The vertical order of the lines in the legend does indeed correspond to their order in the plot. We added this point to the figure caption.

22- We modified the plot to make the OnlineFlow line thicker.

Section 5 23- Thank you for noting this, we fixed the typo.

Acknowledgments 24- Thank you for pointing this out, while the use of tenses was intentional the quotes were not and are now fixed.

References 25- We tried to address the problems to the best of our ability.

---

## Round 1 · Referee Report · Anonymous (Referee 2) · 2022-4-7

Report

This was a very pleasant read. Clearly and pedagogically written, and
the proof-of-concept investigations were straight-forward to
follow. Most of the machine learning tools used are well known and
tested, but putting them in an extreme online environment in this way
is a novel and very interesting idea, which certainly merits a
publication in SciPost.

The only thing I miss in this paper is a more proper feasibility study
for actually implementing this at the LHC. I understand that this is
beyond the scope of the paper, but it would have been nice to see at
least an outlook giving the steps such a feasibility study would need
to follow. Can the authors envisage potential show-stoppers?

As an example I would have liked to see comments about the scalability
of the concept. In the more advanced test case, the authors used only
five input variables per event, and storing those to disk for detailed
off-line analysis would surely not be a problem, even for a year of
LHC running at an event rate of 100 kHz. So, assuming that the authors
envisage a much larger set of input variables, it would be nice to
understand how more complex would the network architecture need to
be. And how much larger the buffer sizes and longer the training
cycles for each buffer. What would this mean in terms of the
requirements for the FPGAs that the author suggest should be
used? Would they then be fast enough to handle the 100 kHz event rate
online?
  • validity: -
  • significance: -
  • originality: -
  • clarity: -
  • formatting: -
  • grammar: -

Author:  Sascha Diefenbacher  on 2022-06-29  [id 2620]

(in reply to Report 2 on 2022-04-07)

We greatly appreciate the feedback. We agree that a full feasibility study would be of interest, however it does indeed exceed the current scope of this work. We did however add further elaboration about the potential difficulties of a full scale implementation of this.

---

## Round 2 · Referee Report · Anonymous (Referee 2) · 2022-7-13

Report

I think the authors successfully addressed all comments from me and the other referee, and recommend that the manuscript is published as is.

---

## Round 2 · Referee Report · Anonymous (Referee 1) · 2022-8-2

Report

The authors successfully addressed and replied to the points of concern. The manuscript should be published.

---

## Round 2 · Author Response

We are grateful to the referees for their input and thoughtful comments and suggestions. We integrated the several suggestions into the text, with any modifications indicated in the list of Changes. Additionally we would like to address some of the raised points in more specific detail:

1- The impact the signal contamination has on the training could be made clearer. For example, one might naively guess that the signal is tiny in the beginning and that each update will make the signal significance larger. Here in lies the confusion. If the signal is larger after the update, won’t the subsequent update absorb the signal in the background model? Some clarification may alleviate such concerns.

-> Thank you for raising this point, however we are unsure about the source of the confusion. The signal fraction remains constant throughout a single training, as it would be purely defined by the underlying physics. The update indicated in Fig.2 is not intended as an automated, continuous process that takes effect during the training, but rather as an after-the-fact cross check in case any hints of new physics are found in the OnlineFlow scheme.

2- As a follow-up to the previous point, can/does OnlineFlow account for changing pileup conditions? Assuming that it does this well, to what order-of-magnitude level of S/B can this tool find new physics? Presumably given enough training cycles and updates it can squeeze out the significance given any S/B.

-> We agree that the exact behavior of the OnlineFlow for various signal rates would be interesting to explore, however we feel this to be beyond the current scope of this work. We added a remark about the behavior for various S/B ratios and how this is an interesting region to explore. As for pileup, the objects that go into the flow would be pileup corrected, so there should be no difference compared to an offline search.

3- The physics case considered is the LHC Olympics dataset containing a W’ decay filtering on having a jet with pT > 1.2 TeV. This feel like an odd choice to showcase as the threshold for the lowest unprescaled single jet trigger at CMS and ATLAS was much lower at around 500 GeV during Run-2. One can imagine an argument that the specifics of this problem does not matter as it is demonstrating the capability and that for deployment lower pT jets would be targeted. Is that true? Assuming that it is, the reader is left to wonder whether the results of Figure 9 still holds. Some clarification would be appreciated.

-> Thank you for noting this, while this dataset features high mass resonances that are not perfectly in line with the intended application range of OnlineFlow, we feel that the proven and well known nature of the LHCO data, as well as its availability make up for this.

4- The role, rationale, and impact of dummy variables is vague. The appearance of dummy variables in the parametric example seems less controversial since standard optimization tools may not work for for 1d inputs for technical reasons. Three is a mystery here, but not of big concern. However, for the LHCO problem the inputs are doubled with dummy variables. What is going on? -> The origin of the three dummy variables stems from doubling the input twice, from 1 to 2 to 4 in total. The addition of the dummy variable in the LHCO setting is indeed not required, however we saw and improvement to the training behavior through their inclusion. This however not systematically explored. We further added more clarification on the origin as use of the dummy variables to the paper

5- The introductory sentence gives the impression that the collision rate might be going up, rather than the trigger rate at the same collision rate. Perhaps there is a way to reword.

-> Thank you for pointing this out, however the purpose of the first sentence is to say that the collision rate is too high to record all events - there is nothing to do with the trigger there. Therefore, we prefer to keep these sentences as they are. The logical flow here is: (1) collision rate is too high to save everything. (2) Need to save partial information, either by throwing out events, throwing out parts of events, or both. Each method has advantages and disadvantages.

6- The last sentence of the penultimate paragraph is confusing. Isn’t the network determination based on training data?

-> While the network is based on the training data, those data are not saved. What we mean by this sentence is that the generative model trained online could be equivalent to saving all of the training data for offline”.

7- Would a sketch or example of the idea of “generative ML” be too pedagogical? Perhaps even a simple definition would help.

-> While we can understand the point, we do feel that a introduction into generative models would not be the best fit for this paper.

8- The bulleted steps is helpful. One part that triggered a question is step 3, where it mentions “indication of new physics”. How can the system distinguish new physics compared to a detector flaw? The authors may know of a paper by A. Pol et al (https://arxiv.org/abs/1808.00911) that discusses detector monitoring using anomaly detection. It would be interesting to know what the authors think of the new physics vs. detector issues ambiguity, even if it is beyond the scope of the paper. If appropriate, please add the citation.

-> We understand how this question arises, however the OnlineFlow setup itself does not aim to distinguish between new and known physics. The principled idea is to store a representation of the measured data while using less effective storage space. How this representation of the data is later on treated in specific is beyond the scope of this work, the two anomaly detection methods used (BumpHunt and CWoLa) serve as a non-exhaustive example. The update to the trigger system is not intended to happen in an automated online setting, but rather involves modifying trigger menus after analyzing the representation of the data in an offline setting.

9- The presentation of where and how OnlineFlow could be deployed could be made more concrete. Only after reading the paper does the reader understand that the tool is to be deployed at HLT-like environments where all of the reconstructed inputs, including sophisticated variables that require tracking information, are made available as inputs. This preprocessing is taken for granted—which may be somewhat reasonable for HLT, although that would depend whether full tracking is needed—and this would not be a given for FPGA/GPU-like setups.

-> We do mention the deployment on HLT as the first proposed deployment level quite early on in this section and we feel that an even earlier mention of this would disrupt the text flow too much.

10- Section 3.1, after referencing figure 4, states that the OnlineFlow reproduces the peak. OnlineFlow seems to have a weak bump above background, but it is much flatter than the S+B. Is that what you mean? This feels like it did not reproduce the peak. Please clarify.

-> As the goal is in finding evidence of new physics in a subsequent offline analysis of the OnlineFlow data, any abundance, even if it does not perfectly model the signal peak, may still be sufficient to find new physics. We have added a sentence to this section detailing this and further elaborating on the signal peak agreement.

11- In section 3.3, it’s not clear to the reader that the parameters given in equation 4 are important enough to document it in the paper. Perhaps it is sufficient to state them in the figure? Also, the relevant figure should be referenced somewhere earlier in section 3.3.

-> We do feel the parameters are important as they allow for comparison between the fit result on the OnlineFlow data and the true underlying function parameters. Stating them in the figure unfortunately resulted in a difficult to read figure. Unfortunately we are not sure which figure is referred to here, as in our eye the fit parameters would not seem appropriate or relevant for the significance comparison in Fig.6. Further Fig.5, which does show a fit, does not use OnlineFlow data, but the analysis performed on the training data, so adding the parameters here does not seem intuitive either.

12- Is there significance of prescale factor ~ 4? It’s not obvious to the reader why 4, if significant. Perhaps it is empirical to the problem at hand. Please clarify.

-> The prescale factor of 4 is not inherently significant, however it is the prescale factor at which the OnlineFlow significance and data significance crossover. To further help that point we specifically indicated the crossover point in the relevant figure.

---

## Round 2 · List of Changes

- Section 1, Paragraph 2: removed superfluous citation
- Section 1, Paragraph 2: changed “the training of sophisticated networks on such devices is still an active area of research” to “the training of graph-based networks on such devices is still an active area of research
- Section 1, Paragraph 2: changed “On the other hand, the all-GPU first trigger stage of LHCb might allow the ready deployment of this idea” to “At the same time, the available resources limit the size and therefore complexity of possible ML models.”
- Section 1, Paragraph 3: added footnote: “as training (as opposed to inference) models on FPGA hardware deployed at earlier trigger stages is currently not possible”
- Section 1, Paragraph 3: changed “However, a sufficiently optimized version of this approach could transform data taking by removing the need for triggers altogether.” to “However, a sufficiently optimized version of this approach could transform data taking by instead learning the overall distribution of data without the need for triggers altogether.”

- Section 2, Paragraph 1: changed “they are hardware-based and do not perform complex reconstruction.” to “they are hardware-based and at most perform low-complexity reconstruction.”
- Section 2, Itemized list: changed “analyse” to “analyze”
- Section 2, Figure 2: separated L1T and HLT into separate boxes and to clarify the input the OnflineFlow receives

- Section 3.1, Paragraph 2: changed “we add three dummy dimensions drawn from a normal
distribution, resulting in a total of four input and output dimensions.“ to “we add three dummy dimensions drawn from a normal distribution, quadrupling the total number of input and output dimensions to four.“
- Section 3.1, Paragraph 3: changed “In Fig. 4 we show how the generated OnlineFlow events, trained on signal plus background” to “In Fig. 4 we show how OnlineFlow, trained on signal plus background”
- Section 3.1, Paragraph 3: added “While the width of the peak is not correctly described we do see a noticeable abundance. As the goal is in finding evidences of new physics in a subsequent offline analysis of the OnlineFlow data, this abundance may still, however, be sufficient for our purpose even if the peak width is mismodeled.”

- Section 3.3, Paragraph 8 (end of section): changed “We see that the OnlineFlow fake rate for the flow varies more than for the training data, but stays well below the signal significance we achieve for the 0.5% signal contamination.” to “We see that there is a larger error margin for the OnlineFlow significance, owing to the larger fluctuations between individual OnlineFlow training, however the average fake rate for the flow stays well below the signal significance we achieve for the 0.5% signal contamination.”
- Section 3.3, Paragraph 8 (end of section): changed “Throughout, the fake rate is at the level of the classical bump hunt.” to “The average also stays consistent with the fake rate of the classical bump hunt.”
- Section 3.3, Paragraph 8 (end of section): added “The precise behavior for intermediate signal rates presents an interesting question that exceeds the scope of this work, but would warrant further investigation.”
- Section 3.3, Figure 6: added grey line indicating the position of the crossover point
- Section 3.3, Figure 6: added in caption “The dotted grey line indicates the crossover point at a datafaction of 1 , which corresponds to a prescale factor of 4.”

- Section 4.1, Paragraph 3: added “While this dataset features high mass resonances that are not perfectly in line with the intended application range of ONLINEFLOW, we feel that the proven and well known nature of the LHCO data, as well as its availability make up for this shortcoming.”
- Section 4.1, Paragraph 6: changed “... , five features and five additional noise dimensions, which we find improves the performance.” to “... .These comprise five features and five additional noise dimensions, the additional noise was found to increase the performance, although no systematic scan over this hyperparameter was performed.”

- Section 4.2, Figure 9: added in caption “Vertical order of the Data lines corresponds to their order in the legend.”
- Section 4.2, Figure 9: increased thickness of OnlineFlow line

- Section 5, Paragraph 4: changed “We believe these examples serve as a prove of concept for the proposed OnlineFlow, warranting further investigation into ways to optimize the setup and into applying it to current trigger systems” to “Implementing OnlineFlow into an existing trigger system will require further work to scale up the networks input dimensionality as well as its expressiveness to handle the more complex data structures of real LHC events. Further, the challenge of integrating the model training into the infrastructure of a real experiment will require further work and exploration. However, regardless of these challenges, we believe the examples demonstrated here serve as a proof of concept for the proposed OnlineFlow, warranting further investigation.”

- Acknowledgments: changed “Quantum Universe" to Quantum Universe

- References: numerous style changes.

---

## Editorial Decision

published